# The Presentation of Two Unrelated Clinical Cases from the Republic of North Ossetia-Alania with the Same Previously Undescribed Variant in the *COL6A2* Gene

**DOI:** 10.3390/ijms232012127

**Published:** 2022-10-12

**Authors:** Sofya A. Ionova, Aysylu F. Murtazina, Inna S. Tebieva, Zalina K. Getoeva, Elena L. Dadali, Polina A. Chausova, Olga A. Shchagina, Andrey V. Marakhonov, Sergey I. Kutsev, Rena A. Zinchenko

**Affiliations:** 1Research Centre for Medical Genetics, Moskvorechie str. 1, 115522 Moscow, Russia; 2North Ossetian State Medical Academy of the Ministry of Health of the Russian Federation, Pushkinskaya str. 40, 362019 Vladikavkaz, Russia; 3Republican Children’s Clinical Hospital, Barbashova str. 33, 362003 Vladikavkaz, Russia; 4Pravoberezhnaya Central District Clinical Hospital, Kominterna str. 12, 363020 Beslan, Russia; 5N.A. Semashko National Research Institute of Public Health, Vorontsovo Pole str. 12-1, 105064 Moscow, Russia

**Keywords:** Republic of North Ossetia-Alania (RNOA), Digors, *COL6A2*, Ullrich congenital muscular dystrophy (UCMD), collagenopathy, high-throughput sequencing (HTS)

## Abstract

Here, we described three affected boys from two unrelated families of Ossetian-Digor origin from the Republic of North Ossetia-Alania who were admitted to the Research Centre for Medical Genetics with unspecified muscular dystrophy. High-throughput sequencing was performed and revealed two novel frameshift variants in the *COL6A2* gene (NM_001849.3) in a heterozygous state each in both cases: c.508_535delinsCTGTGG and c.1659_1660del (case 1) and c.1689del and c.1659_1660del (case 2). In two cases, the same nucleotide variant in the *COL6A2* gene (c.1659_1660del) was observed. We have suggested that the variant c.1659_1660del may be common in the Ossetian-Digor population because two analyzed families have the same ancestry from the same subethnic group of Ossetians). The screening for an asymptomatic carriage of the nucleotide variant c.1659_1660del in 54 healthy donors from Ossetian-Digor population revealed that the estimated carrier frequency is 0.0093 (CI: 0.0002–0.0505), which is high for healthy carriers of the pathogenic variant. Molecular genetic, anamnestic data and clinical examination results allowed us to diagnose Ullrich muscular dystrophy in those affected boys. Genetic heterogeneity and phenotypic diversity of muscular dystrophies complicate diagnosis. It is important to make a differential diagnosis of such conditions and use HTS methods to determine the most accurate diagnosis.

## 1. Introduction

Neuromuscular disorders (NMDs) represent a diverse group of diseases which is characterized by muscle dysfunction due to pathology of the peripheral nervous system, the neuromuscular junction, or skeletal muscle. Clinical presentation includes a wide range of symptoms but commonly include muscle weakness, muscle atrophy, abnormal or impaired ambulation, joint contractures, and skeletal deformities [1]. NMDs vary in terms of age of onset, severity, and prognosis [2]. Hereditary and acquired forms are distinguished. Hereditary NMDs represent a diverse group of genetically heterogeneous diseases with high phenotypic variability. On the one hand, mutations in one gene could lead to several diseases (spinal muscular atrophy, Charcot-Marie-Tooth disease), on the other hand, mutations in several genes also could lead to one or similar diseases [2]. For example, collagen type VI myopathies which are caused by mutations in one of three collagen-VI genes, namely *COL6A1*, *COL6A2*, and *COL6A3*, are associated with several disorders, including more severe Ullrich congenital muscular dystrophy (UCMD) (OMIM #254090) and mild to moderate Bethlem myopathy (BM) (OMIM #158810) [3,4,5].

The Laboratory of Genetic Epidemiology at the Research Centre for Medical Genetics (RCMG) studied the prevalence of hereditary diseases and has identified frequent and rare nosologically forms in various populations and ethnic groups in the Russian Federation [6,7,8,9,10,11,12,13,14]. Since 2017, comprehensive genetic and epidemiological studies of the population of the Republic of North Ossetia-Alania (RNOA) have been performed [11,13]. The RNOA is the republic of the Russian Federation inhabited by more than 20 ethnic groups, one of which is Ossetians. Ossetians are an ethnic group living in the North Caucasus of Russia, which is the autochthonous population of the RNOA. There are several sub-ethnic groups (Irons, Digors, Kudars, Tualians), the first two of which are the most numerous. Two unrelated families from the RNOA were diagnosed in the RCMG. The first family had a long story of diagnosis odyssey because of the wrong primary referral clinical diagnosis. The whole exome sequencing was made for the one proband from the first family, which revealed two novel variants in the *COL6A2* gene. After a while, another family living in another district of the RNOA with an affected male proband with muscular dystrophy was admitted. The next generation sequencing-based gene panel was made for the second proband, which also revealed two novel variants in the *COL6A2* gene, one of which was the same as in the previous family.

Two unrelated families with common nucleotide variant in the *COL6A2* gene were of Ossetian-Digor origin. The screening 54 of healthy inhabitants of Ossetian-Digor origin for an asymptomatic carrier state of c.1659_1660del nucleotide variant in the *COL6A2* gene was performed that allowed us to estimate assumed disease prevalence among Ossetians-Digors.

## 2. Results

### 2.1. Clinical Data

#### 2.1.1. The First Family

The first proband was a 17-year-old boy when he was referred to our center. The patient never walked on his own due to the severe weakness of his leg muscles. He had contractures of the knee and hip joints. It is known from the anamnesis that the one-year-old male patient with flaccid tetraparesis was referred to different pediatric hospitals where he was diagnosed with spinal muscular atrophy (SMA) at one center and congenital undifferentiated myopathy at another one based on clinical data. The subsequent molecular genetic study did not confirm this diagnosis. The proband was born from the first preterm delivery at 36–37 weeks of gestation in breech presentation. The pregnancy proceeded against the background of oligoamnios from six months and the threat of miscarriage at seven months. The child did not cry at birth, had an Apgar score 2/4/7, a low birth weight of 2390 g (−2 SD), a birth length of 52 cm (+1 SD), and a birth head circumference of 35 cm (0 SD). Moreover, he had congenital heart disease, atrial septal defect, hip dysplasia, intrauterine infection of unspecified etiology, and disseminated intravascular coagulation. The baby was attached to artificial pulmonary ventilation, and on the eighth day, he was transferred to the neonatal pathology unit. During the first two years of life, the proband had plaster immobilization of the lower extremities for the hip dislocations, resulting in atrophy of the leg muscles and delayed motor development. The biochemical tests showed increased levels of lactate and creatine phosphokinase (3.3 mmol/L (normal range is 0.5–2.2 mmol/L) and 247 U/L (normal range is 25–200 U/L) respectively) (Table 1). At the age of three years, magnetic resonance imaging (MRI) of the lumbosacral spine revealed a cyst of 60 × 12 mm in size with high-protein content, which compressed the vertebral canal. The needle electromyography (EMG) performed at the age of four years showed the signs of the neurogenic process in leg muscles with moderate spontaneous activity resembling fasciculation potentials, fibrillation potentials, and positive sharp waves.

At the time of neurologic examination at the age of 17, the patient was non-ambulant, had diffuse muscle hypotonia and atrophy, significant limb muscle weakness, spine rigidity, flexion contractures of upper and lower extremities, equinovarus deformity of feet, hypermobility of interphalangeal joints, and tendon areflexia (Figure 1A). Intellectual functions were normal. The healthy parents of the proband do not have consanguineous relations, are Ossetians-Digors by origin.

The patient has two male siblings, and one of them is similarly affected (Figure 1B). The affected brother had marked motor development delay and walked from the age of 5 years. At the time of clinical examination, he was an eight-year-old boy with diffuse muscle atrophy, moderate weakness of the proximal and distal limb muscles, spine rigidity, and flexion contractures of upper and lower extremities. For many years, he was observed with a diagnosis of SMA because of marked motor development delay, diffuse muscle atrophy, and neurogenic signs on needle EMG.

#### 2.1.2. The Second Family

In the second family, the proband was a 14-year-old boy with severe generalized muscle atrophy, muscle weakness, and joint contractures, from a non-consanguineous Ossetian-Digor family. It is known from anamnesis that the boy was born from the third pregnancy by cesarean section with an Apgar score 6/8, birth weight of 3700 g (0 SD), and birth length of 57 cm (+1 SD). During pregnancy, reduced fetal movements were noted. At birth, the baby was transferred to the neonatal pathology unit, where he was diagnosed with craniospinal birth injury, intraventricular hemorrhage, and distal flaccid paraparesis. Starting from the first months, the motor development delay was noted, the proband started walking at 19 months, but his gait was waddling. At the age of four years, he was admitted to a hospital for rehabilitation because of flexion joint contractures of the lower extremities. After four years, he gradually lost the ability to walk due to progressive muscle weakness of the legs. For a long time, he was observed with a diagnosis of SMA, until the age of nine when he underwent needle EMG, which showed a myogenic change without pathologic spontaneous activity in the muscles of upper and lower extremities. There were no signs of peripheral neuropathy or motor neuron lesions.

The neurological examination at the age of 14 years showed diffuse muscle atrophy, prominent weakness of neck muscles, proximal and distal limb muscles, long myopathic face, flexion contractures of the large joints, prominent kyphoscoliosis with asymmetrical chest deformity, lumbar hyperlordosis, and spine rigidity (Figure 1E–G). Some ophthalmological findings were noted as arcuate opacity of the cornea on the right side, and densification of the lens on the left side.

The biochemical tests showed an increased creatine kinase level up to 305 U/L (the normal range is 25–200 U/L) (Table 1). The rest of the biochemical parameters were normal. Parents and 20-year-old female siblings of the proband are healthy (Figure 1D).

Comparative information about the patients’ clinical data is summarized in Table 1.

### 2.2. Genetic Analysis

The first proband initially was searched for deletion of exons 7–8 in the *SMN1* gene, but no changes were found. Therefore, the proband was searched for pathogenic variants using the whole exome sequencing (WES). Two novel variants in the *COL6A2* gene in a heterozygous state each were found: c.508_535delinsCTGTGG in exon 3 and c.1659_1660del in exon 21, both leading to a frameshift and formation of a premature translation termination codon, p.(Cys170LeufsTer20) and p.(Lys554ArgfsTer11), respectively. Sanger sequencing was performed only for the proband and his brother since the biological material of their parents was not available. Both variants in the *COL6A2* gene were validated in a heterozygous state in both brothers (Figure 2A,B).

The second proband with symptoms of neuromuscular pathology was admitted to the RCMG. Proband was searched for pathogenic variants using the next-generation sequencing-based gene panel. Two novel variants in the *COL6A2* gene in a heterozygous state each were found: c.1659_1660del in exon 21 and c.1689del in exon 22, both leading to a frameshift and formation of a premature translation termination codon, p.(Lys554ArgfsTer11), and p.(Gly564ValfsTer32), respectively. Sanger sequencing was performed for the proband and his parents, which confirmed the carrier status of the parents and the compound heterozygous state of the proband (Figure 2A,C).

All novel variants should be considered as probably pathogenic with the level of significance PM2, PVS1 according to the ACMG criteria.

### 2.3. Population Analysis

Single nucleotide variant c.1659_1660del in the *COL6A2* gene was found in two unrelated cases from the RNOA in the heterozygous state. We have hypothesized that this variant could be common in Ossetian-Digor population. The analysis of the carrier frequency of single nucleotide variant c.1659_1660del in the *COL6A2* gene was performed for 54 Ossetians-Digors from the RNOA, which revealed one variant carrier. So, the carrier frequency of variant c.1659_1660del in the *COL6A2* gene among Ossetian-Digor population from the RNOA is 0.0093 (*n* = 1/108) (95% CI: 0.0002–0.0505). Thus, the frequency of heterozygous carriage is 1:54 people (considering 95% CI: 1:10–1:2500 people). Based on these results, the assumed disease prevalence in Ossetian-Digor population in the RNOA has been estimated: the average value was 8.6 people per 100,000 population (considering 95% CI: 0.004–255 per 100,000 population).

## 3. Discussion

Three affected males from two unrelated families from RNOA (two boys from the first family and one boy from the second family) were admitted to RCMG with a clinical picture of muscular dystrophy but without a clear clinical diagnosis.

Thus, NGS was conducted for the probands. In both cases, two novel variants in a heterozygous state were found in the *COL6A2* gene resulting in premature stop-codon formation (for the first proband—c.508_535delinsCTGTGG and c.1659_1660del; and for the second proband–c.1659_1660del and c.1689del) which should lead to mRNA degradation through the nonsense-mediated decay mechanism [15]. Sanger sequencing was performed for two probands and their family members, who are available for genetic analysis. (Figure 2). The affected male sibling for the first family confirmed the presence of both variants of the *COL6A2* gene. Parents from the second family are healthy carriers. Pathogenic variants in the *COL6A2* gene lead to a functional decrease in the α2-chain of type VI collagen (ColVI). ColVI is ubiquitous non-fibrillar collagen composed of three chains: α1(VI), α2(VI), and α3(VI), which interact with each other to assemble in the endoplasmic reticulum and form heterotrimers [16,17,18,19]. They are encoded by three genes, i.e., *COL6A1*, *COL6A2*, and *COL6A3*, respectively. Pathogenic variants affecting these genes lead to different effects on gene expression and protein function. A decrease or absence of ColVI protein leads to the formation of an unstable extracellular matrix that is no longer attached to cells through the basement membrane [17]. As a result, the stability of muscle cells and connective tissue gradually decreased, leading to muscle weakness, contractures, and other signs and symptoms of collagen VI-associated myopathies. These include Bethlem myopathy 1 (BM) and Ullrich congenital muscular myopathy (UCMD) which are thought to represent the ends of a clinical spectrum that also includes intermediate phenotypes of variable severity [16,20,21,22,23,24]. Collagen VI-associated myopathies could be inherited in both autosomal-dominant and autosomal-recessive modes. In the autosomal-dominant mode, the main molecular mechanism is haploinsufficiency, but the dominant-negative effect of missense mutations is also described [20,25]. Biallelic mutations cause complete or profound loss of function of ColVI-genes and are inherited in autosomal recessive mode [5].

UCMD is characterized by generalized muscle weakness and hypermobility of distal joints in conjunction with variable contractures of more proximal joints and normal intelligence [21]. Since birth, UCMD patients present significant muscular weakness and hypotonia. In severe cases, independent movement is not achieved, or loss of independent ambulation occurs in early childhood [26,27,28]. Other typical features of UCMD include progressive scoliosis and deterioration of respiratory function [21,29]. The age of onset of BM is much more variable. BM could be presented with mild developmental motor delay, moderate weakness, and atrophy of the muscles of the trunk and limbs, meanwhile, proximal muscle weakness is more affected in comparison with distal muscles. The progression is slow, and patients reach an advanced age [3,5,30]. In our case, there are two families with the typical manifestations of the UCMD with severe clinical picture: clinical symptoms developed from birth and progressed rapidly with age, two male probands lost the ability to walk an early age, and all boys presented motor development delay, flexion contractures of limb joints, and progressive muscle weakness. In addition, all patients demonstrated signs of different connective tissue damage and the need for artificial pulmonary ventilation: heart disease, atrial septal defect, disseminated intravascular coagulation, hip dysplasia (in the second proband). In both cases, using molecular genetics methods and anamnesis data, Ullrich congenital muscular dystrophy was diagnosed.

The affected sibs from the first family initially were diagnosed with SMA because of the severe generalized muscle atrophy and neurogenic changes on needle EMG. An electromyography pattern in hereditary muscular dystrophies sometimes is misinterpreted as neurogenic because of moderate or severe spontaneous activity and because of the presence of motor unit potentials with very high amplitude usually seen in neurogenic processes [31]. The first proband (older brother from the first family) additionally had a cyst in the lumbosacral region which compressed the vertebral canal, and theoretically could be a cause of neurogenic changes in leg muscles. The needle electromyography of upper limb muscles was not performed on this proband.

In two cases, HTS methods allowed us to determine the most accurate diagnosis. The HTS methods are especially important for severe pathologies with manifestation in early childhood when it is necessary to determine the diagnosis quickly and precisely. Given the clinical polymorphism and genetic heterogeneity of hereditary diseases of the nervous system and the overlapping variability of symptoms, with not always a clear clinical picture, the use of HTS methods plays a leading role in the establishment of the diagnosis.

It is noteworthy that during the genetic analysis the same nucleotide variant in the *COL6A2* gene was found in both unrelated cases. Given the fact that both patients in our cases were Ossetians-Digors from the RNOA, we supposed that single nucleotide variant c.1659_1660del in the gene *COL6A2* may be common in Digor population. So, we analyzed the carrier status of variant c.1659_1660del in the *COL6A2* gene in 54 unrelated healthy inhabitants from Digor population. Only one individual is the carrier, so the carrier frequency of variant c.1659_1660del in population is 0.0093. The high frequency of this variant in the Digor population suggests the possible founder effect, the phenomenon of shifting genetic diversity. The founder effect is probably a result of isolation and subsequent “bottle-neck” effect in the Digor subethnic group of Ossetians.

In the Laboratory of Genetic Epidemiology in the RCMG, a comprehensive medical and genetic examination of the RNOA populations was performed, during which more than 100 nosological forms were described, including neuromuscular diseases, but no collagenopathies were found among them. Our finding may become a reason for the further screening of collagenopathies among populations of other districts of RNOA.

## 4. Materials and Methods

### 4.1. Patients

Three affected male patients from two unrelated families were included in the study. The first family is a proband 17-year-old boy who has an 8-year-old affected brother, and the second family with one proband is a 15-year-old boy. They underwent clinical examination and molecular genetic diagnosis at the RCMG (Moscow, Russia). The parents from the first case were not available for clinical and genetic analyses. The studies involving human participants were reviewed and approved by The Ethical Committee of the Research Centre for Medical Genetics (Protocol No. 5 dated 20 December 2010). All members of the family investigated in the present study signed the informed written consent form (or responsible consent form for infant probands) as anonymous participants of the study and donors of biological materials.

### 4.2. Source of DNA

Peripheral blood samples were collected from patients and their relatives and from 54 unrelated clinically healthy people of Digor origin. All individuals investigated in the present study have signed the informed written consent form as anonymous participants of the study and donors of biological materials. Genomic DNA was obtained using standard procedures from peripheral blood samples.

### 4.3. High-Throughput Sequencing

Whole-exome sequencing has used for the first proband, which was performed using a BGISEQ-500 instrument with average on-target coverage of 146× with MGIEasy Exome Capture V4 kit (BGI, Beijing, China) for library preparation (Genomed Ltd., Moscow, Russia). Bioinformatic analysis was performed using an in-house software pipeline as described earlier [PMID: 29504900]. In brief, it included quality control of raw reads (FastQC tool v. 0.11.5) followed by read mapping to the hg19 human genome assembly (bwa mem v. 0.7.1), sorting of the alignments, and marking duplicates (Picard Toolkit v. 2.18.14). Base recalibration and variant calling were performed with GATK3.8. Variant annotation was done using Annovar tool (v.2018Apr16). Further filtering was performed by functional consequences and population frequencies according to the ACMG recommendations as well as clinical relevance determined by Human Phenotype Ontology database [PMID: 33264411].

For the second proband, next-generation sequencing-based gene panel was analyzed using a new-generation Ion S5™ sequencer (ThermoFisher Scientific, Waltham, MA, USA). The libraries for sequencing were prepared with ultramultiplex PCR based on Ion AmpliSeq™ technology and included 44 genes associated with congenital muscular dystrophies. Sequencing data were processed using a standard automated algorithm in Torrent Suite™ (ThermoFisher Scientific, Waltham, MA, USA) and NGS-DATA software [32]. The detected variants were analyzed using a hg19 genome assembly, HGMD Professional Database v2020.2, and ACMG criteria for variant interpretation [33].

All variants were named according to the reference transcript variant NM_001849.3 of the *COL6A2* gene.

### 4.4. Polymerase Chain Reaction (PCR), Sanger Sequencing

The novel variants in the *COL6A2* gene were confirmed by Sanger sequencing. Table 2 shows the primers used. PCR was performed according to standard protocols.

### 4.5. Population Study

The carrier frequency of single nucleotide variant c.1659_1660del in the COL6A2 gene in the Digor population from RNOA was assessed by heteroduplex analysis in 30% polyacrylamide gel (Figure 3). The frequency was calculated and the assumed prevalence of the disease was estimated using WINPEPI software [34]. The confidence interval (CI) for carrier frequency and prevalence was calculated using the Klopper–Pearson method.

## 5. Conclusions

In this study, firstly, we established the diagnosis in two cases using HTS methods. Secondly, we described three novel variants in the *COL6A2* gene, each of which affects the function of the protein. Moreover, we demonstrated that the nucleotide variant c.1659_1660del in the *COL6A2* gene is quite common among healthy asymptomatic carriers of the Ossetian-Digor population in RNOA, which is undoubtedly important in the diagnosis of neuromuscular diseases. During medical genetic counseling, it is necessary to consider the possible carrier stage of the nucleotide variant c.1659_1660del in the *COL6A2* gene among to Ossetian-Digor population from RNOA.

## Figures and Tables

**Figure 1 ijms-23-12127-f001:**
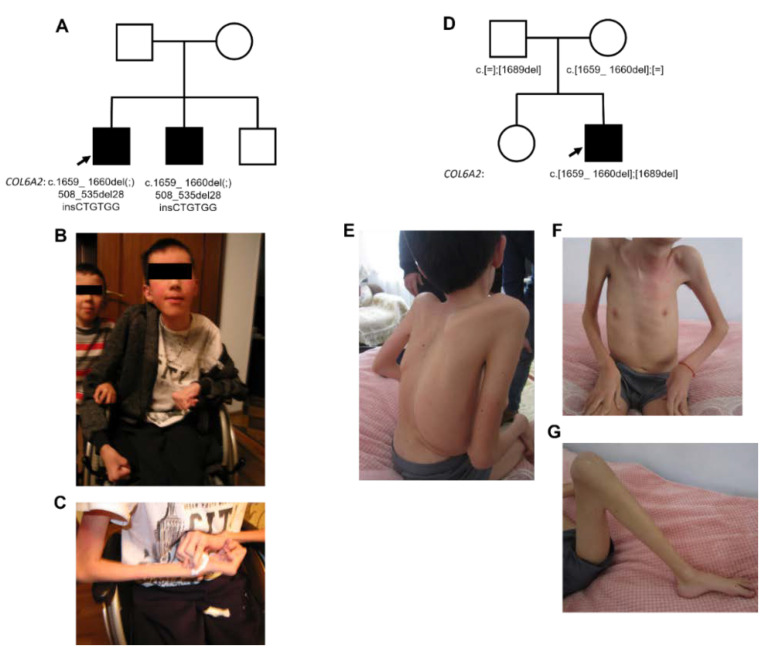
The pedigree of the first family (**A**) and clinical presentation of the first proband: the patient is non-ambulatory (**B**), contractures of the wrist and elbow joints, and prominent diffuse atrophy of upper limb muscles (**B**,**C**). The second family pedigree (**D**) and the clinical presentation of the second proband: diffuse muscle atrophy (**E**–**G**), prominent kyphoscoliosis with asymmetrical chest deformity (**E**,**F**), flexion contractures of the large joints (**F**,**G**).

**Figure 2 ijms-23-12127-f002:**
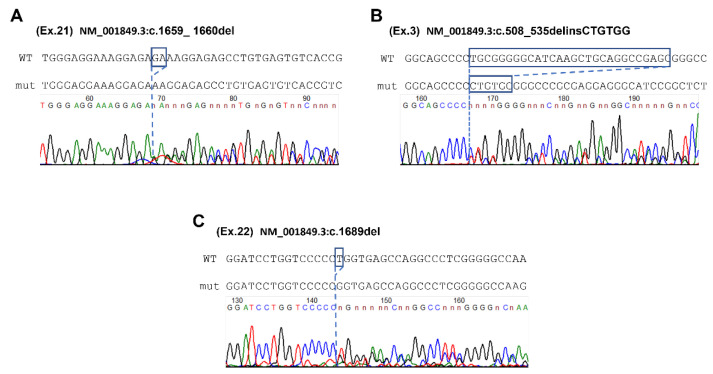
Sanger sequencing of c.1659_1660del variant (spotty blue line indicates deletion site) (**A**), c.508_535delinsCTGTGG variant (spotty blue line indicates deletion sites) (**B**), and c.1689del variant (spotty blue line indicates deletion site) (**C**).

**Figure 3 ijms-23-12127-f003:**
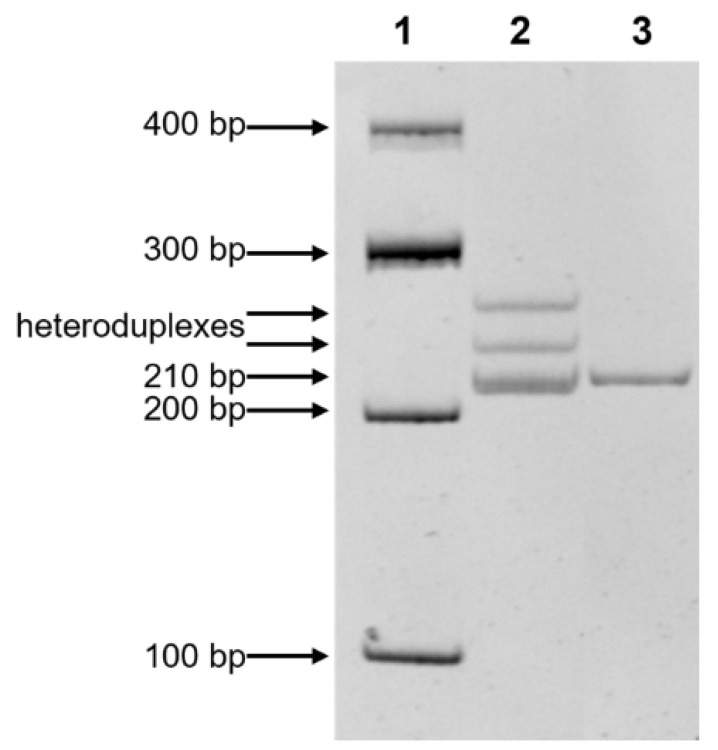
Electrophoregram of PCR-based test system for population screening for c.1659_1660del nucleotide variant in *COL6A2* gene. Lanes: 1—100 bp DNA Ladder; 2—heteroduplexes indicate a heterozygous state (WT/mut); 3—WT/WT (210 bp).

**Table 1 ijms-23-12127-t001:** Comparison table with clinical data.

Patient	1.1	1.2	2.1
Genotype (NM_001849.3)	c.[1659_1660del]; [508_535delinsCTGTGG]	c.[1659_1660del]; [508_535delinsCTGTGG]	c.[1659_1660del]; [1689del]
Ancestry	Ossetians-Digors	Ossetians-Digors	Ossetians-Digors
Age/gender	17 y.o./male	8 y.o./male	14 y.o./male
Hypotonia at birth	+	n/d	+
Apgar score	2/4/7	n/d	6/8
Birth weight	2390 g (−2 SD)	n/d	3700 g (0 SD)
Birth length	52 cm (+1 SD)	n/d	57 cm (+1 SD)
Motor development delay	+	+	+
Ambulance	non ambulant	ambulant	non ambulant
Diffuse muscle atrophy and weakness	+	+	+
Kyphoscoliosis	+	+	+
Spine rigidity	+	+	+
Hip dysplasia	+	+	-
Large joint contractures	+	+	+
Distal joint laxity	+	+	+
Tendon areflexia	+	+	+
Feet deformation	+	+	+
Lactate level (normal: 0.5–2.2 mmol/L)	3.3	n/d	n/d
CK level (normal: 25–200 U/L)	247	n/d	305
Intellectual functions	normal	normal	normal
Additional findings	intrauterine infection at birth	n/d	arcuate opacity of the cornea on the right side, densification of the lens on the left side

**Table 2 ijms-23-12127-t002:** Sequences of primers were used to validate revealed variants.

*COL6A2* Variant	Primer Name	Sequence of Primers, 5′→3′	Position (hg19) and Size of Amplicon
c.508_535 delinsCTGTGG	*COL6A2*-ex3	F: CCACGTCACCGGCAG	chr21:47532266-47532586, 320 bp
		R: CGCATCACAGAGGGGTAAA	
c.1659_1660del	*COL6A2*-ex21	F: CTGGTAGAGACAGCTCCT	chr21:47542738-47542947, 210 bp
		R: TGGCGTTCTCCTGTACTC	
c.1689del	*COL6A2*-ex22	F: ATAGGGAAGGAGGAGGCACAG	chr21:47544404-47544700, 297 bp
		R: AGAACCACGAGGCTTGGGTG	

## Data Availability

The datasets used and/or analyzed during the current study are available from the corresponding author upon reasonable request.

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
