# Peer review of "The Presentation of Two Unrelated Clinical Cases from the Republic of North Ossetia-Alania with the Same Previously Undescribed Variant in the COL6A2 Gene"

_ijms, 2022, doi:10.3390/ijms232012127_

Round 1

Reviewer 1 Report

I have some comments to be considered by the authors:

ABSTRACT

In the background part of the abstract, I suggest to focus on the phenotypic diversity and the genetic heterogeneity of muscular dystrophies which makes the difficulty in diagnosing such conditions.

INTRODUCTION :

RESULTS :

- Please check the birth height of the second proband (57cm ?)

- in figure 1 legends, please add a description of the clinical signs on the different photos

-add the complete terminology for WES

-check the nomenclature of the variant : c.508_535del28insCTGTGG  (the correct for mis : c.508_535delinsCTGTGG)

-in the 15th line of the paragraph genetic analysis (2.2), please correct the sentence « .....p.(Gly564ValfsTer32), and p.(Lys554ArgfsTer11), respectively ». The order should be inverted: p.(Lys554ArgfsTer11) and p.(Gly564ValfsTer32) respectively

- Figure 2 legend in A and C: please correct « spotty line indicates the deletion site » instead of insertion

DISCUSSION

-Line 5 in discussion: please replace « preterm stop-codon » with « premature stop codon »

- line 6 in disuccion : please correct the nomenclature c.510_535del28insCTGTGG

(should be c.508_535delinsCTGTGG)

- in the sentence « in addition, all patients demonstrated signs of different connective tissue damage.... » please add: and the need for artificial pulmonary ventilation

- given the recurrent variant in both families and its frequency in healthy controls from the region of North Ossetia Alania, add a discussion on the possible founder effect of this variant.  

Additional remarks :

-please replace the PMID numbers in the text with the corresponding number from the reference list.

- the article needs a thorough English editing

Author Response

First, we eager to thank the Reviewers for detailed and thorough analysis of our manuscript as well as valuable comments and recommendations to improve it.

Reviewer1

I have some comments to be considered by the authors:

ABSTRACT

In the background part of the abstract, I suggest to focus on the phenotypic diversity and the genetic heterogeneity of muscular dystrophies which makes the difficulty in diagnosing such conditions.

ANSWER

Thank you for your recommendation, in abstract we represented summary of our article. We tried to shortened abstract without losing meaning focusing on the phenotypic diversity and the genetic heterogeneity of muscular dystrophies.

INTRODUCTION :

RESULTS :

- Please check the birth height of the second proband (57cm ?)

Answer: Thank you, we double-checked this parameter in anamnesis and the birth height of the second proband is 57 cm.

- in figure 1 legends, please add a description of the clinical signs on the different photos

Answer: Thank you, we added clinical signs on photos in Figure 1.

-add the complete terminology for WES

Answer: Thank you, we added complete terminology for WES in description of genetic analysis (2.2) (page 4).

-check the nomenclature of the variant : c.508_535del28insCTGTGG  (the correct for mis : c.508_535delinsCTGTGG)

Answer: Thank you, we checked the nomenclature variant and corrected it throughout the manuscript.

-in the 15th line of the paragraph genetic analysis (2.2), please correct the sentence « .....p.(Gly564ValfsTer32), and p.(Lys554ArgfsTer11), respectively ». The order should be inverted: p.(Lys554ArgfsTer11) and p.(Gly564ValfsTer32) respectively

Answer: Thank you, we corrected this mistake.

- Figure 2 legend in A and C: please correct « spotty line indicates the deletion site » instead of insertion

Answer: Thank you, we corrected description in Figure 2.

DISCUSSION

-Line 5 in discussion: please replace « preterm stop-codon » with « premature stop codon »

Answer: Thank you, we checked incorrect spelling « preterm stop-codon » throughout the article and corrected it.

- line 6 in discuccion : please correct the nomenclature c.510_535del28insCTGTGG

(should be c.508_535delinsCTGTGG)

Answer: Thank you, we checked incorrect spelling throughout the article and corrected it.

- in the sentence « in addition, all patients demonstrated signs of different connective tissue damage....» please add: and the need for artificial pulmonary ventilation

Answer: Thank you, we added «and the need for artificial pulmonary ventilation» in sentence.

- given the recurrent variant in both families and its frequency in healthy controls from the region of North Ossetia Alania, add a discussion on the possible founder effect of this variant. 

Answer: Thank you for your recommendation. We supposed the founder effect, and it is important to say about it. So, we changed the discussion and added sentence about possible founder effect.

Additional remarks:

-please replace the PMID numbers in the text with the corresponding number from the reference list.

Answer: Thank you, we replaced PMID numbers.

- the article needs a thorough English editing

Answer: Thank you for your recommendation, we edited English thorough the article.

Reviewer 2 Report

The manuscript entitled ‘The presentation of two unrelated clinical cases from the Republic of North Ossetia-Alania with the same previously undescribed variant in the COL6A2 gene’ by Sofya A. Ionova et al. represents an interesting manuscript about a novel variant of COL6A2 gene. However, the present manuscript form is very preliminary and more experiments are needed for the patient’s characterization.

Even clinical and genetic characterization could be improved with an introduction of a summary table.  Parameters like Max motor ability (Power grade), Contractures, Distal laxity, creatine kinase level, Respiratory function, cardiac function, Collagen VI in SKM, or others that the authors considered important should be included.

The immunohistochemistry of muscle biopsies for collagen VI are very important. Further, the patient-derived skin fibroblasts can also be used to evaluate the expression levels of Collagen VI and protein levels.

As mentioned before, the present form of the manuscript should not be accepted for publication in the IJMS.

Author Response

First, we eager to thank the Reviewers for detailed and thorough analysis of our manuscript as well as valuable comments and recommendations to improve it.

Reviewer2

The manuscript entitled ‘The presentation of two unrelated clinical cases from the Republic of North Ossetia-Alania with the same previously undescribed variant in the COL6A2 gene’ by Sofya A. Ionova et al. represents an interesting manuscript about a novel variant of COL6A2 gene. However, the present manuscript form is very preliminary and more experiments are needed for the patient’s characterization.

Even clinical and genetic characterization could be improved with an introduction of a summary table.  Parameters like Max motor ability (Power grade), Contractures, Distal laxity, creatine kinase level, Respiratory function, cardiac function, Collagen VI in SKM, or others that the authors considered important should be included.

The immunohistochemistry of muscle biopsies for collagen VI are very important. Further, the patient-derived skin fibroblasts can also be used to evaluate the expression levels of Collagen VI and protein levels. As mentioned before, the present form of the manuscript should not be accepted for publication in the IJMS.

Answer: Thank you for your valuable advice and recommendations. We added a summary table with clinical signs with many parameters, parameters of muscular dystrophy and other information. Analysis of collagen expression in skin fibroblasts would be important and significant, but, unfortunately, we do not have an opportunity to perform it because the biological material from the patients is unavailable. Also, we do not have the possibility to perform immunohistochemistry analysis on muscle biopsies data. We consider your advice and will try to evaluate the protein level.

Reviewer 3 Report

This manuscript by Sofya A. Ionova et al reported Col6A2 mutations that caused unspecified muscular dystrophy. They used whole-exome sequencing and sanger sequencing identified deletion in exon 3, exon 21, 22 in two different families. These new discovered mutations can guide diagnose of these kinds of genetic disease. The manuscript is well written and fit the topic of the special issue.   

Author Response

First, we eager to thank the Reviewer for detailed and thorough analysis of our manuscript as well as valuable comments and recommendations to improve it.

Reviewer 3

This manuscript by Sofya A. Ionova et al reported Col6A2 mutations that caused unspecified muscular dystrophy. They used whole-exome sequencing and sanger sequencing identified deletion in exon 3, exon 21, 22 in two different families. These new discovered mutations can guide diagnose of these kinds of genetic disease. The manuscript is well written and fit the topic of the special issue.  

Answer: Thank you very much for your opinion on our article.

Round 2

Reviewer 2 Report

The manuscript entitled ‘The presentation of two unrelated clinical cases from the Republic of North Ossetia-Alania with the same previously undescribed variant in the COL6A2 gene’ by Sofya A. Ionova et al. represents an interesting manuscript about a novel variant of COL6A2 gene. However, the present manuscript form is very preliminary and more experiments are needed for the patient’s characterization.

From my previous suggestions they just added a summary table with clinical signs with some parameters, parameters of muscular dystrophy and other relevant information.

The authors did not added more experiments, therefore the present form of the manuscript should not be accepted for publication in the IJMS.